# A sampling survey of enterococci within pasteurized, fermented dairy products and their virulence and antibiotic resistance properties

Solomon H. Mariam ●*

Aklilu Lemma Institute of Pathobiology, Armauer Hansen Research Institute, Addis Ababa University, Addis Ababa, Ethiopia

* solomon.habtemariam@aau.edu.et

**Data Availability Statement:** All relevant data are together with the paper and its Supporting Information files.

**Funding:** The author received no specific funding for this work.

## Abstract

Globally, fermented foods (FFs), which may be traditional or industrially-produced, are major sources of nutrition. In the traditional practice, the fermentation process is driven by communities of virtually uncharacterized microflora indigenous to the food substrate. Some of these flora can have virulent or antibiotic resistance properties, posing risk to consumers. Others, such as *Enterococcus faecalis* and *Enterococcus faecium*, may also be found in such foods. Enterococci that harbor antibiotic resistance or virulence factors can cycle among animals, food, humans and the environment, thereby transferring these harmful properties at the gene level to harmless commensals in the food matrix, animals and humans. In this work, several microbial isolates obtained from different FF sources were analyzed for their identity and virulence and/or antibiotic resistance properties. For identification aiming at enterococci, isolates that were Gram-positive and catalase- and oxidase-negative were subjected to multiple tests including for growth in broth containing 6.5% NaCl, growth and hydrolytic activity on medium containing bile-esculin, hemolytic activity on blood agar, and growth at 45˚C and survival after incubation at 60˚C for 30 min. Furthermore, the isolates were tested for susceptibility/resistance to a select group of antibiotics. Finally, the isolates were molecularly-characterized with respect to species identity and presence of virulence-encoding genes by amplification of target genes. Most sources contained enterococci, in addition to most of them also containing Gram-negative flora. Most of these also harbored virulence factors. Several isolates were also antibiotic-resistant. These results strongly suggest attention should be given to better control presence of such potentially pathogenic species.

## Introduction

An association between humans and microbes begins early in life, and eventually the gastrointestinal tract (GIT) alone becomes host for numerous species of microbes (collectively called

**Competing interests:** The author declares that no competing interests exist.

the gut microbiota), but other body sites also represent important niches [1,2]. A large portion of this pool of microbes is yet uncultivated, or uncultivable. Consequently, this pool remains partially uncharacterized with respect to their overall roles in health and disease, and many aspects (e.g., antimicrobials, virulence, etc) remain largely yet to be explored [3]. Consumption of food products of fermentation may serve as a source of live lactic acid bacteria (including species that reportedly provide human health benefits) and may influence the diversity and composition of the gut microbiota [4,5]. This pool may be subject to consistent flux of intake and exit, and colonization may be transient due to encountered stress (although some may be resilient). However, even transient colonization may have negative impact (e.g., antibiotic resistance transfer) [6]. Survival of yogurt bacteria during passage through the GIT is considered a prerequisite for colonization. Several studies [7–14] indicate that much of the yogurt bacteria indeed survive the stress (e.g., acid, bile) encountered during passage.

There are great diversities in the types of fermented food (FF) products as well as their microbial contents and numbers present within [4]. Modern or industrial food fermentation practices introduced defined starter culture systems that, unlike the traditional ones, appear to be better characterized with respect to their identities and other characteristics [15]. Such fermentations in most other countries often are spontaneous fermentations, relying on indigenous microbes present in the starting raw materials [16].

Lactic acid bacteria (LAB) (such as those of the genera *Lactobacillus*, *Lactococcus*, *Streptococcus*, and *Enterococcus* are among the microbial flora responsible for food fermentations. Probiotics (live microorganisms which when administered in adequate amounts confer health benefit on the host) are deliberately added to processed foods and promoted as healthful food additives. However, although probiotics have been used for centuries and appear to be safe for some people, there are still controversies regarding the probiotic-derived benefit or side effects. Some [e.g., 17,18] warn of possible side effects, or that the verdict on probiotic safety is still pending. Others [e.g., 19] grossly and indiscriminately advocate probiotic inclusions in foods, a premature recommendation. Even probiotics that have long established use may have side effects. A case in point is that of *Escherichia coli* Nissle 1917, a strain that has been in use as a probiotic for more than a century to alleviate intestinal disorders such as ulcerative colitis and Crohn's disease [20]. This strain was shown to cause damage (genotoxic effect) in the form of double-strand breaks in DNA mediated by possession of a peptide-polyketide-encoding genomic island that was also shown to be inseparable from its probiotic activity [20,21], although this disentanglement may be possible through genetic manipulation for its probiotic effect devoid of genotoxicity [22]. Perhaps, the future is for engineered probiotics.

Enterococci can be found in FFs as contaminants, primarily because of their resistance to disinfectants and adaptation to harsh environments [23,24]. They also have capacity to share antibiotic resistance and other determinants. Enterococci of both clinical and food origin possess genomes of high plasticity due to abundance of pheromone-responsive and broad host range plasmids, enabling them to colonize a wide range of niches and habitats and thereby both acquire and spread genetic traits, including resistance and virulence genes [25,26]. Acquisition and spread of antibiotic resistance in enterococci can be mediated via mobile genetic elements such as resistance plasmids, transposons and integrative conjugative elements [27,28]. Intra- or inter-species gene exchange, resulting in sharing or transfer of antibiotic resistance, altered metabolic capabilities, and other virulence factors (VFs) has been observed in enterococci in *in vitro* and *in vivo* conditions [29,30]. For example, transfer of *vanA* gene has been reported [6,31,32]. Furthermore, enterococci exhibit intrinsic resistance to several clinically-relevant antibiotics (such as *β*-lactams, aminoglycosides, lincosamides, and cephalosporins). Additionally, enterococci acquire resistance through mutations, enhancing their power as causes of patient morbidity and mortality.

Although enterococci have been perceived as having low pathogenic potential, this is changing as enterococci are opportunistic pathogens and some are emerging as major pathologic agents and nosocomial pathogens as a result of several factors including their possession of VFs and pheromone-responsive plasmid-gene-transfer mechanisms. Thus, they can be found as colonizers in several infections (especially those caused by multidrug-resistant strains in patients hospitalized for prolonged periods or with underlying diseases) including urinary tract infections, endocarditis and sepsis (thought to be due to dissemination from the gut into the bloodstream or along intravenous lines [33]. As with many other infectious diseases, treatment of infections caused by enterococci is mostly empirical. This can have consequences including failed treatment outcome, morbidity, mortality and succession of the gut commensal flora with flora of different composition (such as the pathogenic *Clostridium* family), or vancomycin-resistant enterococci [34].

The multiple VFs that enterococci express include aggregation substance (*agg*), gelatinase (*gelE*), cytolysin (*cyl*), enterococcal surface protein (*esp*), accessory colonization factor (*ace*), hyaluronidase (*hyl*), enterococcal endocarditis antigen (*efaA*) and sex pheromones (e.g., *cpd*, *cob*, *ccf*, *cad*). Aggregation substance is a surface protein encoded by sex pheromone plasmids that binds to extra-cellular matrix proteins of eukaryotic cells and promotes donor and recipient cell aggregation and conjugative transfer of plasmid virulence traits and antibiotic resistance genes [35,36]. *gelE* codes for a protein toxin with gelatinolytic activity capable of hydrolyzing gelatin, collagen, hemoglobin, and other bioactive compounds. Cytolysin, encoded by the *cyl* operon, is composed of several cytolysin genes (*cylM*, *cylB*, *cylA*, etc) that may be involved in transport, activation, modification and expression of cytolysins (combines activities of both its own expression and lytic activity against a broad range of prokaryotic and eukaryotic cells) [37]. *esp*, which encodes for a cell wall-associated protein on the bacterial surface, is involved in immune evasion (due to expression of alternate forms of the Esp protein), and is mostly found in infection-derived *E. faecalis* isolates [38]. The enterococcal accessory colonization proteins (Ace in *E. faecalis* and Acm in *E. faecium*) are collagen and laminin adhesins thought to be involved in endocarditis, as both *ace* and *acm* deletion mutants are attenuated in experimental endocarditis in rats, and antibodies specific for the collagen-binding domain of Ace protect rats from endocarditis [39]. $efaA_{fs}$ was originally isolated from serum of an endocarditis patient, and its homolog in *E. faecium* ($efaA_{fm}$) subsequently identified [40,41]. Many of these VFs appear to be transmissible via pheromone-responsive conjugative plasmids [42,43].

The high diversity of the microbiota and their intimate association with states of health and/or disease engenders the question as to what constitutes a healthy or unhealthy microbiota? Since a germ-free state is neither attainable nor sustainable in life [44,45] and at least some of the gut microbiota come from consumption of food products of fermentation, it is therefore imperative that, probiotic or not, microbes that are acquired from such foods are characterized for pathogenic potential. In Ethiopia, information on virulence properties and possible health impacts of *Enterococcus* (a major food-borne pathogen) in such foods is scarce or nonexistent. In fact, it was not even possible to find any validated figures regarding the share of FFs in Ethiopian staple food. Even a National Food Consumption Survey prepared by the Ethiopian Public Health Institution did not adequately mention the share of dairy products to the national diet, although it is known that FFs in general and dairy products in particular account for a large share. The objectives of this small-scale study were to conduct assessment of both the prevalence and possible virulence traits of enterococci that exist within FFs of pasteurized dairy origin that are extensively consumed by the population, in order to provide a glimpse of the situation at the wider scale.

## Materials and methods

### The study site

Addis Ababa is the capital of Ethiopia. The city's milk supplies come mainly from dairy industries and cooperatives. The same brands of dairy products are distributed for sale at all supermarkets of/in the city. The population size is estimated at ~4 million. Moreover, it has a significant international presence, being the seat of many diplomatic, continental and global organizations.

### Isolation and presumptive identification of bacteria

Broth and agar media used for isolation and growth of bacteria were from Becton Dickinson (Sparks, Maryland) or Oxoid (Basingstoke, Hampshire, United Kingdom). For isolation of bacteria, yogurt, cheese and milk purchased from supermarkets were used as source of bacteria. Ten mL samples of each yogurt brand were serially-diluted in physiological saline and dilutions plated onto MRS, plate count or tryptic soy agars. Single colonies were picked and inoculated into MRS and/or brain heart infusion (BHI) broth and incubated. Mid-log phase cultures were pelleted to store the pellet, while portions were also inoculated onto agar media. From this growth on agar media, Gram-stains were made to select for Gram-positive cocci. Culture media were from Becton Dickinson (BD) and Oxoid. Culture media used were MRS agar or broth, M17 agar, tryptic soy agar, BHI broth, blood agar and PCA agar (for total LAB count).

Colonies were re-streaked 3 successive times, each from 24-hr-grown agar cultures of single colonies pre-grown in broth, to get pure isolates (until the Gram stain showed no mix of Gram-negatives and Gram-positives or cocci and rods). The tests were aimed at isolating enterococci. Isolates that were Gram-negative or with yeast appearance were saved but not further characterized.

### Catalase test

Catalase test was conducted by adding 2–3 drops of 3% $H_2O_2$ on cultures.

### Oxidase test

The oxidase test was performed using Oxidase test strips (OXOID, Australia) by adding 1–2 drops of oxidase reagent on sterile petri dishes smeared with a loopful of the bacterial growth. Once pure isolates (Gram-positive, catalase- and oxidase-negative) were obtained, further tests were conducted.

### Growth in 6.5% NaCl

Isolates were incubated in BHI broth containing 6.5% NaCl at 37˚C for 24–48 h. All isolates were also simultaneously and similarly incubated in BHI broth without NaCl.

### Esculin hydrolysis test

This was conducted by streak plating onto bile esculin (BE) agar plates and incubating at 37˚C for 24 hrs. Isolates were also simultaneously inoculated onto agar without BE.

### Growth tests at high or low temperatures

Tests for survival after incubation at 60˚C for 30 minutes in a water bath were conducted using diluted broth cultures pre-grown for 24 hrs. CFU were quantified both before and after

incubation. Growth tests at 45˚C were conducted by incubating broth cultures at 45˚C for 24 or 48 h. Growth tests at 10˚C were conducted by incubating broth cultures at 10˚C for 24 or 48 h.

## Hemolysis test

This was conducted by inoculation of Columbia blood agar base containing 5% defibrinated sheep blood and incubating at 37˚C for 24 hrs.

The biochemical, salt tolerance and temperature tests were performed following standard procedures [46].

## Antibiotic susceptibility tests (ASTs)

Preliminary ASTs were conducted for some of the isolates by disk diffusion using Muller-Hinton (MH) agar. Bacterial suspensions were prepared in physiological saline (0.85%) from 18-hr-grown agar cultures to closely match the density of MacFarland Standard 0.5. These were then spread on the MH agar on petridishes to make a thick paste using sterile cotton swabs and antibiotic-impregnated disks (Abtek, England) were placed on the agar after 15 minutes of inoculation. The disk contents in μg were erythromycin, 15; tetracycline, 30; ampicillin, 10; oxacillin, 1; ciprofloxacin, 5; azithromycin, 15; vancomycin, 5. The plates were placed in 37˚C incubator within 15 minutes of disk placement. Interpretations of results were made from the categories according to the Clinical Laboratory Standards Institute (CLSI) version 2017 [47] (however, not all interpretive criteria were available from CLSI for the available disk contents). *Escherichia coli* ATCC 25922) and *Staphylococcus aureus* ATCC 25923 and *E. faecalis* ATCC 29212 were used simultaneously for quality control in the disk diffusion tests. A limited number of isolates were tested for susceptibility to eight antibiotics using a plate broth microdilution assay (VetMIC Lact2, version 2014–06) (National Laboratory institute, Upsalla, Sweden). All isolates were tested for presence of *ermB*, *ermC* and/or *vanC1*, *vanC2/C3* resistance genes with primers listed in Table 1.

## DNA extraction, polymerase chain reaction (PCR), electrophoresis and capture of gel bands

DNA was extracted using Qiagen Blood and Tissue Kit or by heat treatment for 10 min at 95˚C. The concentrations of the extracts were measured using NanoDrop 2000 (Thermo Scientific). Species-specific PCR was used to identify *Enterococcus faecalis* and *Enterococus faecium* [48]. The primers, product sizes and reference sources are given in Table 1. Modifications on some of the primer concentrations were made. PCR products were separated in agarose gel. Separated bands were viewed and captured using a gel scanner and UV Transilluminator (Bio-Rad, USA).

## Tests for presence of virulence genes

The isolates verified as enterococci by PCR were then submitted for extended virulence content analysis. The virulence genes tested for the presence of included *gelE*, *asa1*, *cylA*, *cylM*, *cylB*, *esp*, *hyl*, *efaAfs*, *efaAfm*, *cob*, *cpd*, *ccf*, and *erm* and *van* genes (Table 1). For the virulence factors *asa1*, *gelE*, *esp*, *hyl* and *cyl*, a multiplex PCR protocol was conducted [49]. The *hyl* gene was not amplified by this protocol, including from *E. faecalis* ATCC 29212 itself. It was not possible to detect the *cob*, *cpd*, and *ccf* genes in a single multiplex PCR reaction, despite several attempts, because *cob* would not amplify with the other two even when it is present. Thus, PCR for *cob* was run separately, but *cpd* and *ccf* were amplifiable together, using optimized

**Table 1. PCR primer sequences and their targets used in this work.**

| Target/Gene | Primer Sequence (5′→3′) | Product size (bp) | Reference(s) |
|---|---|---|---|
| E. faecalis* | ATCAAGTACAGTTAGTCTTTATTAGACGATTCAAAGCTAACTGAATCAGT | 941 | 48 |
| E. faecium* | TTGAGGCAGACCAGATTGACGTATGACAGCGACTCCGATTCC | 658 | |
| asa1 | GCACGCTATTACGAACTATGATAAGAAAGAACATCACCACGA | 375 | 49 |
| gelE | TATGACAATGCTTTTTGGGATAGATGCACCCGAAATAATATA | 213 | |
| esp | AGATTTCATCTTTGATTCTTGGAATTGATTCTTTAGCATCTGG | 510 | |
| hyl | ACAGAAGAGCTGCAGGAAATGGACTGACGTCCAAGTTTCCAA | 276 | |
| cylA | ACTCGGGGATTGATAGGCGCTGCTAAAGCTGCGCTT | 688 | |
| cylB | ATTCCTACCTATGTTCTGTTAAATAAACTCTTCTTTTCCAAC | 843 | 50 |
| cylM | CTGATGGAAAGAAGATAGTATTGAGTTGGTCTGATTACATTT | 742 | |
| efaAfs | GACAGACCCTCACGAATAAGTTCATCATGCTGTAGTA | 705 | |
| efaAfm | AACAGATCCGCATGAATACATTTCATCATCTGATAGTA | 735 | |
| cpd | TGGTGGGTTATTTTTCAATTCTACGGCTCTGGCTTACTA | 782 | |
| cob | AACATTCAGCAAACAAAGCTTGTCATAAAGAGTGGTCAT | 1,405 | |
| ccf | GGGAATTGAGTAGTGAAGAAGAGCCGCTAAAATCGGTAAAAT | 543 | |
| vanA | GGAAAACGACAATTGCGTACAATGCGGCCGTTA | 732 | |
| vanB | ATGGGAAGCCGATAGTCGATTTCGTTCCTCGACC | 635 | |
| vanC-1 | GGTATCAAGGAAACCTCCTTCCGCCATCATAGCT | 822 | |
| vanC2/C3 | CTCCTACGATTCTCTTGCGAGCAAGACCTTTAAG | 439 | |
| ermB | CATTTAACGACGAAACTGGCGGAACATCTGTGGTATGGCG | 405 | 52 |
| ermC | ATCTTTGAAATCGGCTCAGGCAAACCCGTATTCCACGATT | 283 | |

• The primer pairs for *E. faecalis* and *E. faecium* are species-specific.

PCR protocols. Similarly, the reaction product for *efaA*, which like *cob* had to be run separately, was not detected using a previously described protocol [50], but was amplified using a different PCR protocol [51]. For *ermB and ermC* detection, designed primers [52] were used with optimized PCR protocol.

## Data analysis

The presence of whether VFs are present in significant differences based on species or food type were tested using the "t" test for the former and Wilcoxon Rank Sum test for the latter.

## Quality control

*Enterococcus faecalis* 29212, *Pseudomonas aeruginosa* 27853 and *Escherichia coli* 25922, *Streptococcus pyogenes* 19615, *Staphylococcus aureus* 29213, all ATCC strains, were used as positive or negative quality control strains, as relevant, in the various tests. Each experiment was repeated two or three times.

## Results

Altogether, about 70 isolates were isolated from the fermented food products. S1 Table shows the products used in the study (type, size, purchase and expiry dates). S2 Table lists all the isolates (enterococcal, non-enterococcal and others) along with the results of the various tests. In most cases, multiple isolates of various types were obtained from each food product. Several cultures from cheese samples (isolates 34–37, product 4) all contained yeasts as viewed in the Gram stains, with no bacteria. It seemed the presence yeasts was detrimental to bacterial presence. Isolates 39–43 from a different cheese product also contained yeasts.

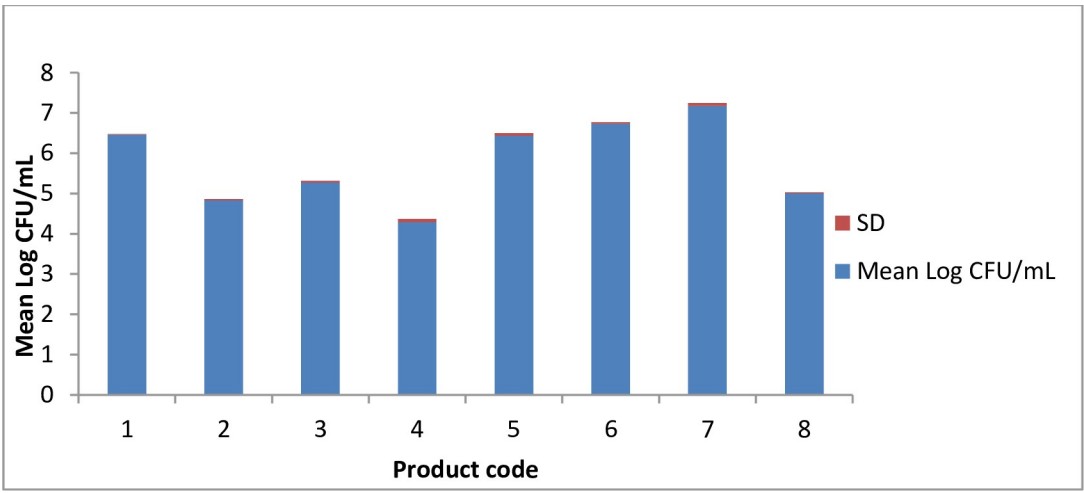

**Fig 1. Microbial loads found in the different products of this study.**

Fig 1 shows the bacterial loads in the different FF products (these are total bacterial counts, not any specific species).

Isolates that were Gram-negative and catalase- and oxidase-positive were excluded from most further tests. Yeasts were also excluded. Isolates were presumptively considered to be enterococci based on the following findings: Gram-positivity and catalase-, and oxidase-nega-tivity, growth in medium containing 6.5% NaCl, growth at temperatures of 45°C for 24–48 hrs and 60°C for 30 minutes. Most isolates grew in culture broth containing 6.5% NaCl and during incubation at 45°C. Isolates 3, 4, 5, 30 and the ATCC *E. faecalis* itself showed growth at 10°C after storage for up to 4 days. Others did not show visible growth after 4 days at 10°C, but grew overnight when left at room temperature after removal from 10°C.

The results of these tests also suggested which isolates may be non-enterococcal LAB (nE-LAB). The study isolates confirmed as enterococci by PCR were isolates 1, 2, 21, 22, 25, 30, 45, 51, and 61–63 (all *E. faecium*) and 3–6 and 52 (all *E. faecalis*) (S2 Table and S1A Fig).

Isolates 3–6 that were found to be *E. faecalis* were independent isolates from a commercial yogurt (Product 7) that is on sale in many supermarkets of the city labeled as "probiotic yogurt". This product also contained the highest bacterial load (Fig 1). Moreover, known VF genes (*gelE*, *cylA*, *cylB*, *cylM*, *asa*, *efaA*, *cpd*, *ccf*, and *cob)* were contained in these isolates (Table 2 and S1B–S1E Fig). The *hyl* gene was not detected in these or the other enterococcal isolates, and the *esp* gene was found in only two (52 and 61) of the isolates tested (Table 2). *efaA* was the most frequently occurring virulence gene, and was found in almost all *E. faecalis* and *E. faecium* isolates. Isolates from cheese, milk and the national collection, most of them *E. faecium*, also contained fewer numbers of VF genes (Table 2). With the caveat that the sample sizes are small, the *E. faecalis* food isolates were found to contain significantly more VF genes than the *E. faecium* isolates (Table 3). With respect to food types, yogurt samples contained the highest number of VFs.

The disk diffusion AST results showed that isolates 2, 3 and 21 were all resistant to six of seven antibiotics while isolates 30 and 31 were both susceptible to vancomycin, but 31 was sus-ceptible to tetracycline as well (S3 Table). Isolates 2 and 3 were also resistant to all antibiotics in the VetMIC format, including to ampicillin and vancomycin (S4 Table). Not all isolates were tested by both disk diffusion and the VetMIC assay (due to limitations in availability), but those tested by the latter assay were all resistant to all antibiotics. Almost all isolates tested

**Table 2. Virulence genes detected in enterococcal isolates of this study.**

| Source/isolate | Virulence factor gene(s) detected | | | | | | | | | | | antibiotic resistance genes | | | |
|---|---|---|---|---|---|---|---|---|---|---|---|---|---|---|---|
| | *gelE* | *esp* | *cylA* | *cylM* | *cylB* | *hyl* | *asa1* | *efaA* | *ccf* | *cpd* | *cob* | *ermB* | *ermC* | *vanC1* | *vanC2/C3* |
| **Yogurt** | | | | | | | | | | | | | | | |
| *E. faecalis* | + | – | + | + | + | – | + | + | + | + | + | – | + | + | – |
| 1* | – | – | – | – | – | – | – | + | – | + | – | – | – | – | – |
| 2* | – | – | – | – | – | – | – | + | + | – | – | + | – | – | – |
| 3** | + | – | + | – | – | – | + | + | + | + | + | – | + | – | + |
| 4** | + | – | + | + | + | – | + | + | + | + | + | – | – | – | – |
| 5** | + | – | + | + | + | – | + | + | + | + | + | + | – | – | + |
| 6** | + | – | – | – | – | – | + | + | + | + | + | – | – | + | – |
| **Cheese** | | | | | | | | | | | | | | | |
| 21* | + | – | – | – | – | – | + | + | – | + | + | – | – | – | – |
| 22* | – | – | – | – | – | – | – | + | – | + | – | – | – | – | – |
| 25* | – | – | – | – | – | – | – | + | + | + | – | – | – | – | – |
| 30* | – | – | – | – | – | – | – | + | + | + | – | – | + | – | – |
| **Milk** | | | | | | | | | | | | | | | |
| 45* | – | – | – | – | – | – | – | – | + | – | – | – | – | – | + |
| 51* | – | – | – | – | – | – | – | + | – | + | + | – | – | – | + |
| 52** | + | + | – | – | – | – | – | + | + | + | + | – | + | – | + |
| **National Collection** | | | | | | | | | | | | | | | |
| 61* | + | + | + | – | – | – | – | + | + | + | – | – | – | – | – |
| 62* | + | – | – | – | – | – | – | – | + | – | – | + | – | – | – |
| 63* | – | – | – | – | – | – | – | + | + | + | – | – | – | – | – |

*—*E. faecium.*

**—*E. faecalis.*

by both methods are considered multidrug-resistant. The VetMIC assay also showed that the five isolates (2, 3, 4, 5 and 32) were resistant to not only vancomycin (MIC $\leq$ 4 μg/mL, CLSI breakpoint), but also to linezolid (S4 Table). Some resistance marker genes for vancomycin ((*vanC1*, *vanC2/C3*) and/or erythromycin (*ermB*, and *ermC*) were also found in several of the isolates (Table 2).

## Discussion

In this study, there were many more non-enterococcal and Gram-negative species in the products. Importantly, many of the tested products that were declared to have been pasteurized

**Table 3. Prevalence of virulence factors (VFs) per species and food type.**

| | No species | Total VFs | Mean | SD | S$^2$ | Variance ratio | t | tc | *P* |
|---|---|---|---|---|---|---|---|---|---|
| A. Virulence factor by species | | | | | | 0.78 | 5.27 | 2.447 | <0.002 |
| *E. faecalis* | 6 | 37 | 7.4 | 1.5 | 2.25 | | | | |
| *E. faecium* | 9 | 18 | 3.33 | 1.323 | 1.75 | | | | |
| | No/food type | Total VFs | Rank sum | | U stat | *P* | | | |
| B. Virulence factor by food type | | | | | U > Uc | < 0.05 | | | |
| Yogurt | 6 | 35 | 38 | | | | | | |
| Cheese | 4 | 13 | 17 | | | | | | |

contained enterococci. The possible sources of enterococci in the products, though not known certainly. could include the environment, dairy workers, animals, and/or the equipments used. The prevalence of enterococci reported here is probably an underestimate, since colony selection was at random and were identified as enterococci after isolation of pure colonies. Currently, no information is available about the types, numbers, and manners of selection of LAB used in the fermentation processes of these commercial products. Nor are there any indications that regulations exist for the control of quality and safety of all commercial fermented dairy products.

The commercial yogurt that is widely on sale (product 7) and that contained enterococci is labeled as both "Pasteurized" and "Probiotic". The general public is unfamiliar with "probiotics"; however, "pasteurized" is in everyone's concept and this gives consumers a sense of security, which is false. Moreover, although the producer declared on the label the microbial flora "present" in the product (3 in total, all of them suppose to be Gram-positive), *Enterococcus* is not one of them. Additionally, all four independent enterococcal isolates contained most of the virulence-encoding genes. The product also contained Gram-negative species. Thus, this product is not what the producer declares it to be. The presence of enterococci in the other "pasteurized" dairy products also suggests that the prevalence of enterococci is widespread. It is possible that other FF products similarly contain enterococci and pathogenic microflora. In general, microorganisms, including *Enterococcus*, can coexist with other bacteria in humans, in foods or culture media [53–55]. In this study itself, most products contained enterococci along with other Gram-positive and/or Gram-negative bacteria.

Foods of various sorts (dairy, poultry, seafood, etc) are major vehicles that transmit pathogenic bacteria that carry virulence or toxin-encoding genes. For example, enterotoxin- and virulence-encoding genes were found in methicillin-resistant *S. aureus* strains isolated from subclinical bovine mastitis [56]. Similarly, enterotoxin gene was detected in methicillin-resistant *S. aureus* isolates from chicken [57]. These can also involve multi-drug resistant Gram-negative pathogens [58–60]. Multidrug-resistant infections are generally more prevalent in nosocomial settings, but are spreading into communities. These have multifaceted effects including worse outcomes, increased morbidity and mortality, increased use of broad-spectrum antibiotics (thereby exacerbating the drug resistance problem and eventually exhausting treatment options) and exaggerated healthcare costs [61,62]. Foremost among the involved pathogens are methicillin-resistant *Staphylococcus aureus*, extended spectrum $\beta$-lactamse-producing enterobacteriaceae, and vancomycin-resistant enterococci. A link between antibiotic use in food animals and rates of antibiotic resistance in humans is known to exist, as exemplified by the use of avoparcin (a glycopeptide) in Europe in the 1990's, with the consequent rise in vancomycin-resistant enterococci [63].

A recent guidance by the European Food Safety Authority (EFSA) [64] detailed that any *E. faecium* strain with ampicillin MIC $> 2$ μg mL$^{-1}$ and in possession of the genetic elements IS*16*, *hyl*Efm and *esp* should be considered unsafe. Conversely, those strains lacking these elements are to be considered safe as feed additives. The presence of these markers has been reported to be positively-correlated with clinical, epidemic or hospital-associated strains of *E. faecium* [38,65,66]. Unless it can be ascertained that strains that lack the above elements also consequently lack the other known enterococcal VFs, it seems doubtful that strains lacking the described genetic elements would be safe. In fact, some [67] argue that the EFSA designation does not guarantee a safe strain and *E. faecium* strains without those elements can be associated with severe human infections. Most isolates tested here were resistant to ampicillin.

Enterococci display intrinsic resistance to $\beta$-lactams (via expression of penicillin-binding proteins [PBPs] with low affinity to $\beta$-lactams) and aminoglycosides (AGs) (due to inability of AGs to reach the target ribosome). Enterococci also display acquired resistance to $\beta$-lactams

(via acquisition of *β*-lactamases or mutations in PBPs) and AGs (via acquisition of mobile genetic elements, or enzymatic modifications of hydroxyl moieties of AGs, or ribosomal mutations) [24]. In this study, resistance to the AGs was found to be high, and these antibiotics may be of little if any use against these isolates. However, such AGs may be of use when combined with *β*-lactams (i.e., combination with cell wall active agents), resulting in synergistic bactericidal activity [68]. A limitation of this approach is toxicity resulting from prolonged use of AGs [69]. Most of this study's isolates also displayed resistance to the *β*-lactams. Vancomycin found use as a treatment of choice against aminoglycoside-resistant enterococci, but resistance to it has also emerged world-wide. Other antibiotics (e.g., daptomycin, linezolid, tigecycline) were introduced to deal with van-comycin-resistant enterococci, but enterococcal resistance to each of these drugs also rapidly developed [24]. The tested isolates are considered MDR, including their resistance to linezolid, which was recommended to treat vancomycin-resistant enterococci.

## Limitations

More detailed analyses of FF products with respect to antibiotic resistance and virulence are needed to elucidate the full impact of the findings of this study on public health. Full identification and analyses for possible virulence and antibiotic resistance properties (including possibly transferable resistance) of the non-enterococcal LAB/microbial species, which were not addressed here, are also required.

## Conclusions

Enterococci have been used for food fermentation and preservation, and they were for long considered harmless. However, enterococci also cause opportunistic infections, which earned them the unclassified status for some time as "friend or foe" [70,71]. On top of their possession of both intrinsic and acquired antibiotic resistance, coexistence with other bacteria in various habitats and niches (e.g., foods, animals and humans), evolution in both their resistance and virulence traits, their propensity to acquire and disseminate resistance and virulence, and their involvement in several types of infections (e.g., urinary tract infections, endocarditis, sepsis, etc) make them formidable pathogens [33,72,73].

The findings here strongly indicate there are serious issues of safety of dairy foods. Many of the dairy products tested contained enterococci, along with other Gram-positive and–negative bacterial isolates. There is no clear information about the sources of these microbial flora (resident or contaminants), but it is very likely to be both. The virulence and antibiotic resistance traits associated with these isolates are serious public health concerns. It is very likely that these health risks are actually causing harm to consumers (but these are not recognized), due to gaps in knowledge of cause-effect relationships. These also represent diagnostic challenges (thus misdiagnosis is very likely in these scenarios), which could lead to inappropriate treatment or failure to treat. Elucidating the roles of these and other microbes in metabolism, symbiotic association with the host, and overall role in host resistance/susceptibility to specific epidemic or pandemic bacterial and viral pathogens are goals worth pursuing.

## Supporting information

**S1 Fig.** PCR results for *Enterococcus* identification (A) and VF detection (B, *efaA*; C, *cpd*, *ccf*; D and E, *cob*). PCR for species identification of *Enterococcus*. Upper panel: Lane 1: Molecular weight marker (MWM) (1 Kb Plus), 2: Isolate 1, 3: Isolate 21, 4: Isolate 22, 5: Isolate 2, 6: Isolate 62, 7: Isolate 63, 8: Isolate 51, 9: Isolate 52, 10: Isolate 3, 11: Isolate 4, 12: Isolate 5. Lower panel: Lane 1: MWM (1 Kb Plus), 2: Isolate 61, 3: Isolate 45, 4: Isolate 6, 5: Isolate 25, 6: Isolate 30, 7: *E. faecalis* (ATCC), 8: Isolate 32 (a cheese isolate), 9: Negative control. Isolates 45, 25, and 30

were run for the species PCR again with increased primer/template combinations and found to be *E. faecium* (data not shown). (B) PCR for VF detection (*efaA*). Upper panel: Lane 1: MWM (1 Kb Plus), 2: *E. faecalis* (ATCC), 3: Isolate 1, 4: Isolate 2. 5: Isolate 3, 6: Isolate 4, 7: Isolate 5, 8: Isolate 6, 9: Isolate 21, 10: Isolate 22: 11: Isolate 25, 12: Isolate 30. Lower panel: Lane 1: MWM (1 Kb Plus), 2: *E. faecalis* (ATCC), 3: Blank lane, 4: Isolate 45, 5: Isolate 51, 6: Isolate 52, 7: Isolate 61, 8: Isolate 62, 9: Isolate 63, 10: Negative control. Isolates 51, 62 and 63 were found to contain *efaA* in a different optimized PCR run. (C) PCR for VF detection (*ccf* and *cpd*). Upper panel: Lane 1: MWM (1 Kb Plus), 2: *E. faecalis* (ATCC) (with both *cpd* and *ccf*). 3: Isolate 1, 4: Isolate 2, 5: Isolate 3, 6: Isolate 4, 7: Isolate 5, 8: Isolate 6, 9: Isolate 21, 10: Isolate 22: 11: Isolate 25, 12: Isolate 30. Lower panel: Lane 1: MWM (1 Kb Plus), 2: *E. faecalis* (ATCC), 3: Blank lane, 4: Isolate 45, 5: Isolate 51, 6: Isolate 52, 7: Isolate 61, 8: Isolate 62, 9: Isolate 63, 10: Negative control. Isolates 61 and 63 were found to contain both *ccf* and *cpd* in a different PCR. (D) PCR for VF detection (*cob*). Upper panel: Lane 1: MWM (1 Kb Plus), 2: *E. faecalis* (ATCC), 5: Isolate 4, 6: Isolate 5. Lower panel: Lane 1: MWM (1 Kb Plus), Lane 6: Isolate 62. (E) PCR for VF detection (*cob*) for additional isolates. Upper panel: Lane 1: MWM (1 Kb Plus), Lane 9: Isolate 51, Lane 10: Isolate 52. Lower panel: Lane 1: MWM (1 Kb Plus), Lane 3: Isolate3, Lane 4: Isolate 6. The other lanes in D and E were also loaded with samples but were negative for *cob*.
(PDF)

**S1 Table. Product information.**
(PDF)

**S2 Table. Phenotypic and genotypic test results of bacterial isolates.**
(PDF)

**S3 Table. Disk diffusion test results for the study isolates.**
(PDF)

**S4 Table. Susceptibility test results for selected isolates using VetMIC plate.**
(PDF)

## Acknowledgments

The availability of AHRI facilities for this work is gratefully acknowledged.

## Author Contributions

**Conceptualization:** Solomon H. Mariam.

**Data curation:** Solomon H. Mariam.

**Formal analysis:** Solomon H. Mariam.

**Investigation:** Solomon H. Mariam.

**Methodology:** Solomon H. Mariam.

**Writing – original draft:** Solomon H. Mariam.

**Writing – review & editing:** Solomon H. Mariam.

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
