## [Decision Letter · Decision Letter 0]

3 Jun 2021

PONE-D-21-14580

A survey of types of microbial flora within pasteurized, fermented dairy products, with focus on prevalence and virulence of Enterococci

PLOS ONE

Dear Dr. Mariam,

Thank you for submitting your manuscript to PLOS ONE. After careful consideration, we feel that it has merit but does not fully meet PLOS ONE’s publication criteria as it currently stands. Therefore, we invite you to submit a revised version of the manuscript that addresses the points raised during the review process.

 A major revision is neededThe manuscript should be revised for English editing.

We look forward to receiving your revised manuscript.

Kind regards,

Abdelazeem Mohamed Algammal, Prof, Ph.D

Academic Editor

PLOS ONE

Journal Requirements:

2. We understand that you purchased dairy products from local supermarkets for this study. In your Methods section, please provide additional regarding the source of this material. Please provide the geographic coordinates and names of the purchase locations (e.g., stores, markets), if available, as well as any further details about the purchased items (e.g., lot number, source origin, description of appearance) to ensure reproducibility of the analyses.

Reviewers' comments:

Reviewer's Responses to Questions

**Comments to the Author**

1. Is the manuscript technically sound, and do the data support the conclusions?

Reviewer #1: Yes

2. Has the statistical analysis been performed appropriately and rigorously? 

Reviewer #1: Yes

3. Have the authors made all data underlying the findings in their manuscript fully available?

Reviewer #1: Yes

4. Is the manuscript presented in an intelligible fashion and written in standard English?

Reviewer #1: Yes

5. Review Comments to the Author

Reviewer #1: Comments to authors:

- The current study is very interesting; however the authors should address the below-outlined comments to improve the manuscript quality:

- The manuscript should be revised for English editing by a native English speaker.

- Write Enterococci in correct form all over the manuscript (should be italic)

Title:

I think the work would benefit from the title that contains the main conclusion of the study (should be derived from the conclusion), please modify the title.

Abstract:

- The abstract must illustrate the used methods and the most prevalent results; please give more hints about methods and the most significant results. Beside, rephrase the main conclusion to sound better.

Introduction:

-The introduction needs to be more informative:

-The authors should illustrate the public health importance concerning the emergence of multidrug-resistant (MDR) bacterial pathogens, and the transmission of MDR-Pathogens to humans either by food chain or direct contact with infected animals. Several studies proved the widespread MDR- bacterial pathogens;

The multidrug resistance has been increased globally that is considered a public health threat. Several previous investigations revealed the emergence of multidrug-resistant bacterial pathogens from different origins that could be transmitted to humans either by food chain or direct contact with the infected animal. You should cite the following valuable studies:

1- PMID: 33177849 ; https://doi.org/10.2147/IDR.S276975

2-PMID: 32497922 ; https://pubmed.ncbi.nlm.nih.gov/32497922/

3-PMID: 32235800 DOI: 10.3390/pathogens9030238

4-PMID: 32397408 DOI: 10.3390/pathogens9050362

5-PMID: 32532070 DOI: 10.3390/toxins12060383

6-PMID: 32994450 DOI: 10.1038/s41598-020-72264-4

7-PMID: 31797067 DOI: 10.1186/s13568-019-0920-4

8-El-Sayed M, Algammal A, Abouel-Atta M, Mabrok M, Emam A. Pathogenicity, genetic typing, and antibiotic sensitivity of Vibrio alginolyticus isolated from Oreochromis niloticus and Tilapia zillii. Rev. de Med. Vet. 2019, 4-6;170:80-86.

-The authors should explain the pathogenicity and virulence factors of Enterococci.

-Rephrase the aim of work to be cleared and better sound.

Methods

- Isolation of bacteria from fermented dairy products:

•Add the name of company, and country of the used media.

•Add a specific reference for the isolation of bacteria.

- Add this subtitle: The identification of isolated bacteria:

•Add specific references to the used biochemical reactions.

-Antibiotic susceptibility tests (ASTs):

-Enumerate the names of the involved antimicrobial agents and their disc concentrations.

-PCR: The primers targeting E. faecalis and E. faecium, did you mean 16sr RNA?

-Data analysis: Add the data concerning the used software.

-Results

- Please illustrate the results of antimicrobial susceptibility testing in a table.

-Please illustrate the prevalence of MDR- strains in a table if present.

-Provide the PCR Figures as a supplementary data.

-Discussion

-The discussion is very poor and the authors are advised to illustrate the real impact of their findings.

-Conclusion

-Rephrase your conclusion to sound better: A real conclusion should focus on the question or claim you articulated in your study, whose resolution has been the main objective of your paper? That question now needs to be re-invoked and definitively answered. More still, you need to leave your reader with a higher level of insight into your topic.

6. PLOS authors have the option to publish the peer review history of their article (what does this mean?). If published, this will include your full peer review and any attached files.

Reviewer #1: No

---

## [Author Response · Author response to Decision Letter 0]

25 Jun 2021

Response to Reviewers

1. Please ensure that your manuscript meets PLOS ONE's style requirements, including those for file naming. Response: I have edited the manuscript following the PLOS ONE style requirements. If there are still some things not accordingly, I apologize and will be ready for further modifications.

2. We understand that you purchased dairy products from local supermarkets for this study. In your Methods section, please provide additional regarding the source of this material. Please provide the geographic coordinates and names of the purchase locations (e.g., stores, markets), if available, as well as any further details about the purchased items (e.g., lot number, source origin, description of appearance) to ensure reproducibility of the analyses.

Response: S1 Table gives available information on the dairy products purchased for the study, and includes product type (cheese, milk, yogurt), product size/volume, date of purchase and expiry date indicated on product at time of purchase. It is not possible to mention names of stores where the products were purchased from because the stores can raise legal issues (e.g., defamation) if made public. If necessary, the names can be revealed to the Journal only. 

3. We suggest you thoroughly copyedit your manuscript for language usage, spelling, and grammar. 

Response: I asked a native English speaker to edit the manuscript for English. He edited it and his modification are included in the “Tracked” manuscript version (those with rearrangements, rephrasing, etc). However, he was not willing to be named. Other modifications in the version were made by the author to cut on unnecessary details or better clarify statements. 

• A clean copy of the edited manuscript (uploaded as the new *manuscript* file) Response: It was not possible to make use of a professional editing service due to lack of foreign currency. However, a colleague assisted in the editing of the manuscript (please see 3 above). Both a tracked version showing the changes made (“Revised Manuscript with Tracked Changes”) and an unmarked version of the tracked manuscript (“Manuscript”) have been submitted to PLOS ONE.

4. PLOS ONE now requires that authors provide the original uncropped and unadjusted images underlying all blot or gel results reported in a submission’s figures or Supporting Information files. This policy and the journal’s other requirements for blot/gel reporting and figure preparation are described in detail athttps://journals.plos.org/plosone/s/figures#loc-blot-and-gel-reporting-requirements and https://journals.plos.org/plosone/s/figures#loc-preparing-figures-from-image-files. When you submit your revised manuscript, please ensure that your figures adhere fully to these guidelines and provide the original underlying images for all blot or gel data reported in your submission. See the following link for instructions on providing the original image data: https://journals.plos.org/plosone/s/figures#loc-original-images-for-blots-and-gels.

Response: The original gel shots without any modifications for Supplementary Fig 1A, B, D and E of the original submission have been included with this revised submission. These are the original gel shots that had to be exported to JPEG to be visible. They are now in PDF format labeled “S1_raw_images”. Unfortunately, it was not possible to get the original, unmodified version of original Supplementary Fig 1C. Only the edited version, which is the same as the one submitted initially, is available.

Response: The captions for the Supporting information files have been included at the end of the manuscript. These include two new tables (“S1 Table, Product Information” and “S3 Table, AST Results”) and the original Supplementary Table1 now “S2 Table Phenotypic and genotypic test results of bacterial isolates”) and S1_raw_images (the original Suppl. Fig. 1). 

 5. Review Comments to the Author

Reviewer #1: Comments to authors:

- The manuscript should be revised for English editing by a native English speaker.

Response: I asked a native English speaker to edit the manuscript for English. He edited it and his modifications are included in the “Tracked” manuscript version (those with rearrangements, rephrasing, etc) and also visible in the final version modified with the “Accept” command. Other modifications in the version were made by the author to cut on unnecessary details or to better clarify statements. 

- Write Enterococci in correct form all over the manuscript (should be italic)

Response. Throughout the manuscript Enterococcus has been italicized.

Title:

I think the work would benefit from the title that contains the main conclusion of the study (should be derived from the conclusion), please modify the title.

Response: The title has been modified accordingly to “A sampling survey of Enterococci within pasteurized, fermented dairy products and their virulence and antibiotic resistance properties”

Abstract:

- The abstract must illustrate the used methods and the most prevalent results; please give more hints about methods and the most significant results. Beside, rephrase the main conclusion to sound better.

Response: The abstract has been modified to include more details of the methods and results obtained. 

Introduction:

-The introduction needs to be more informative: Response: The introduction has now been modified. It includes more on Enterococcus virulence, probiotic perils, antibiotic resistance and the rationale for conducting this study.

-The authors should illustrate the public health importance concerning the emergence of multidrug-resistant (MDR) bacterial pathogens, and the transmission of MDR-Pathogens to humans either by food chain or direct contact with infected animals. 

The multidrug resistance has been increased globally that is considered a public health threat. Several previous investigations revealed the emergence of multidrug-resistant bacterial pathogens from different origins that could be transmitted to humans either by food chain or direct contact with the infected animal. You should cite the following valuable studies:

Response: These articles have been cited at relevant paragraphs in Discussion (references 56-60 and 62).

-The authors should explain the pathogenicity and virulence factors of Enterococci.

-Rephrase the aim of work to be cleared and better sound.

Response: More information on Enterococcus pathogenicity, virulence and antibiotic resistance has been added both in the Introduction and Discussion.

Methods

- Isolation of bacteria from fermented dairy products:

•Add the name of company, and country of the used media.

Response: The sources of the culture media have been included.

•Add a specific reference for the isolation of bacteria.

Response: No specific reference was used for isolation. Instead, it relied on routine serial dilution method followed by culture on media commonly used for enterococci and then cultural and biochemical characterization until colonies with typical characteristics were obtained for further characterization by PCR. The major limitation to using specific references for isolation of enterococci was safety. The use of enterococcal selective media (Slanetz and Bartley, Enterococcus Selective agars are recommended. But since these contain sodium azide which is a safety risk, these media were not used in isolation by following published methods.

- Add this subtitle: The identification of isolated bacteria: Response: This has been added together with isolation as Isolation and presumptive identification of bacteria. The molecular identification is also mentioned under “DNA extraction, polymerase chain reaction (PCR), electrophoresis and capture of gel bands”.

•Add specific references to the used biochemical reactions. Response: The reference for the biochemical test procedures have been included (The Microbiology of Drinking Water (2012) – Part 5, Environment Agency, Government of UK)

-Antibiotic susceptibility tests (ASTs):

-Enumerate the names of the involved antimicrobial agents and their disc concentrations. Response: The antibiotic disks and their contents have been inserted under the title “Antibiotic susceptibility tests” in Materials and Methods.

-PCR: The primers targeting E. faecalis and E. faecium, did you mean 16sr RNA? Response: The primers did not target 16S rRNA, they were specific to E. faecalis and E. faecium

-Data analysis: Add the data concerning the used software. Response: The sample size was small, no software was used and it was done manually.

-Results

- Please illustrate the results of antimicrobial susceptibility testing in a table. Response: Antibiotic susceptibility tests were performed for only a limited number of isolates due to shortage of supplies. A few tests were done by broth dilution plates. The results for the limited number of isolates have been submitted in supporting information as S3 Table.

-Please illustrate the prevalence of MDR- strains in a table if present. Response: All of the few tested isolates are MDR, and this is shown in S3 Table and indicated in Results.

-Provide the PCR Figures as a supplementary data. Response: These have been provided in original, unedited form as S1_raw_images in S1 Fig in supporting information.

-Discussion

-The discussion is very poor and the authors are advised to illustrate the real impact of their findings. Response: Indeed this is also very well noted. The Discussion has been expanded to address issues of pathogenicity, and resistance in this revised version.

-Conclusion

-Rephrase your conclusion to sound better: A real conclusion should focus on the question or claim you articulated in your study, whose resolution has been the main objective of your paper? That question now needs to be re-invoked and definitively answered. More still, you need to leave your reader with a higher level of insight into your topic. Response: The Conclusion has been rewritten with modifications to show and better inform readers about the impacts focusing on virulence and antibiotic resistance to go in line with the title and objective.

6. PLOS authors have the option to publish the peer review history of their article (what does this mean?). If published, this will include your full peer review and any attached files.

Response: I agree.

While revising your submission, please upload your figure files to the Preflight Analysis and Conversion Engine (PACE) digital diagnostic tool, https://pacev2.apexcovantage.com/. PACE helps ensure that figures meet PLOS requirements. To use PACE, you must first register as a user. Registration is free. Then, login and navigate to the UPLOAD tab, where you will find detailed instructions on how to use the tool. If you encounter any issues or have any questions when using PACE, please email PLOS at figures@plos.org. Please note that Supporting Information files do not need this step. Response: This has been performed and PACE used to test the figure for conformity.

---

## [Editor Report · Decision Letter 1]

28 Jun 2021

A sampling survey of Enterococci  within pasteurized, fermented dairy products and their  virulence and antibiotic resistance properties

PONE-D-21-14580R1

Dear Dr. Mariam,

We’re pleased to inform you that your manuscript has been judged scientifically suitable for publication and will be formally accepted for publication once it meets all outstanding technical requirements.

Kind regards,

Abdelazeem Mohamed Algammal, Prof, Ph.D

Academic Editor

PLOS ONE